# Mesenchymal stromal cell aging impairs the self-organizing capacity of lung alveolar epithelial stem cells

Diptiman Chanda[1]*, Mohammad Rehan[2], Samuel R Smith[1], Kevin G Dsouza[1], Yong Wang[1], Karen Bernard[1], Deepali Kurundkar[1], Vinayak Memula[1,3], Kyoko Kojima[4], James A Mobley[5], Gloria A Benavides[6], Victor Darley-Usmar[6], Young-iL Kim[1,7], Jaroslaw W Zmijewski[1], Jessy S Deshane[1], Stijn De Langhe[1], Victor J Thannickal[2]*

[1]Division of Pulmonary, Allergy, and Critical Care Medicine, Department of Medicine, Birmingham, United States; [2]John W. Deming Department of Medicine, Tulane University School of Medicine, New Orleans, United States; [3]Department of Surgery, Birmingham, United States; [4]Comprehensive Cancer Center Mass Spectrometry & Proteomics Shared Facility, Birmingham, United States; [5]Department of Anesthesiology and Perioperative Medicine, Birmingham, United States; [6]Department of Pathology, Birmingham, United States; [7]Division of Preventive Medicine, Department of Medicine; University of Alabama at Birmingham, Birmingham, United States

**Abstract** Multicellular organisms maintain structure and function of tissues/organs through emergent, self-organizing behavior. In this report, we demonstrate a critical role for lung mesenchymal stromal cell (L-MSC) aging in determining the capacity to form three-dimensional organoids or 'alveolospheres' with type 2 alveolar epithelial cells (AEC2s). In contrast to L-MSCs from aged mice, young L-MSCs support the efficient formation of alveolospheres when co-cultured with young or aged AEC2s. Aged L-MSCs demonstrated features of cellular senescence, altered bioenergetics, and a senescence-associated secretory profile (SASP). The reactive oxygen species generating enzyme, NADPH oxidase 4 (Nox4), was highly activated in aged L-MSCs and Nox4 downregulation was sufficient to, at least partially, reverse this age-related energy deficit, while restoring the self-organizing capacity of alveolospheres. Together, these data indicate a critical role for cellular bioenergetics and redox homeostasis in an organoid model of self-organization and support the concept of thermodynamic entropy in aging biology.

*For correspondence:
dchanda@uab.edu (DC);
vthannickal@tulane.edu (VJT)

Competing interest: The authors declare that no competing interests exist.

## Introduction

Substantial progress has been made in our understanding of the biology of aging, and these advances have the potential to improve both healthspan and lifespan, while alleviating the burden of age-related diseases. Self-organization in biological systems is a process by which cells reduce their internal entropy and maintain order within these dynamic, self-renewing systems (*Chatterjee et al., 2017*; *Kiss et al., 2009*). Exhaustion of stem/progenitor cells, cellular senescence, and altered intercellular communication have been proposed as aging hallmarks that increase susceptibility to age-related disorders (*Schultz and Sinclair, 2016*). However, the interactions between these hallmarks and whether cellular bioenergetics associated with cellular senescence may account for age-associated stem cell dysfunction and altered cell-cell communication have not been well defined. In this study, we utilized an organoid model to study stem cell behavior and intercellular communication that may

**eLife digest** Many tissues in the body are capable of regenerating by replacing defective or worn-out cells with new ones. This process relies heavily on stem cells, which are precursor cells that lack a set role in the body and can develop into different types of cells under the right conditions. Tissues often have their own pool of stem cells that they use to replenish damaged cells. But as we age, this regeneration process becomes less effective.

Many of our organs, such as the lungs, are lined with epithelial cells. These cells form a protective barrier, controlling what substances get in and out of the tissue. Alveoli are parts of the lungs that allow oxygen and carbon dioxide to move between the blood and the air in the lungs. And alveoli rely on an effective epithelial cell lining to work properly.

To replenish these epithelial cells, alveoli have pockets, in which a type of epithelial cell, known as AEC2, lives. These cells can serve as stem cells, developing into a different type of cell under the right conditions. To work properly, AEC2 cells require close interactions with another type of cell called L-MSC, which supports the maintenance of other cells and also has the ability to differentiate into several other cell types. Both cell types can be found close together in these stem cell pockets. So far, it has been unclear how aging affects how these cells work together to replenish the epithelial lining of the alveoli.

To investigate, Chanda et al. probed AEC2s and L-MSCs in the alveoli of young and old mice. The researchers collected both cell types from young (2-3 months) and aged (22-24 months) mice. Various combinations of these cells were grown to form 3D structures, mimicking how the cells grow in the lungs.

Young L-MSCs formed normal 3D structures with both young and aged AEC2 cells. But aged L-MSCs developed abnormal, loose structures with AEC2 cells (both young and old cells). Aged L-MSCs were found to have higher levels of an enzyme (called Nox4) that produces oxidants and other 'pro-aging' factors, compared to young L-MSCs. However, reducing Nox4 levels in aged L-MSCs allowed these cells to form normal 3D structures with young AEC2 cells, but not aged AEC2 cells.

These findings highlight the varying effects specific stem cells have, and how their behaviour is affected by pro-aging factors. Moreover, the pro-aging enzyme Nox4 shows potential as a therapeutic target – downregulating its activity may reverse critical effects of aging in cells.

account for age-related phenotypes; through these studies, we identify cellular bioenergetics and redox imbalance as critical drivers of these inter-dependent aging hallmarks.

## Results and discussion

The mammalian lung serves an essential role in organismal metabolism, uniquely by serving as the primary organ for systemic exchange of oxygen for carbon dioxide. Regeneration and maintenance of structure-function of the lung are dependent on adult, tissue-resident stem cells that reside in unique niches along the airways (*Basil et al., 2020*; *Hogan et al., 2014*). Type 2 alveolar epithelial cells (AEC2s) serve as facultative stem/progenitor cells in adult mammalian lungs and differentiate into type 1 alveolar epithelial cells (AEC1s) in response to diverse injuries to reconstitute and reestablish the alveolar gas exchange surface (*Barkauskas et al., 2013*). AEC2 regenerative capacity declines with age, resulting in impaired lung injury repair responses, thus increasing susceptibility to various lung diseases (*Schulte et al., 2019*; *Watson et al., 2020*). To further explore the relationship between AECs and lung mesenchymal stromal cells (L-MSCs), we developed an alveolosphere assay system that has been traditionally used to assess AEC2 stemness or regenerative potential (*Barkauskas et al., 2013*). In this assay system, L-MSCs and AEC2s are mixed together in a ratio of 100,000–5000,, respectively, and seeded in Matrigel (*Figure 1A*); alveolospheres with a single layer of epithelial cells composed of both AEC2s (surfactant protein C [SFTPC]-positive) and AEC1s (lung type I integral membrane glycoprotein [T1α]-positive) surrounding a hollow sphere typically form 9–12 days following co-culture (*Figure 1B and C*). L-MSCs expressing platelet-derived growth factor receptor-α (PDGFRα), with a fewer number expressing α-smooth muscle actin (α-SMA), were found primarily in cells lining the outer edges of alveolospheres (*Figure 1D and E*). AEC2s within alveolospheres stained for both the

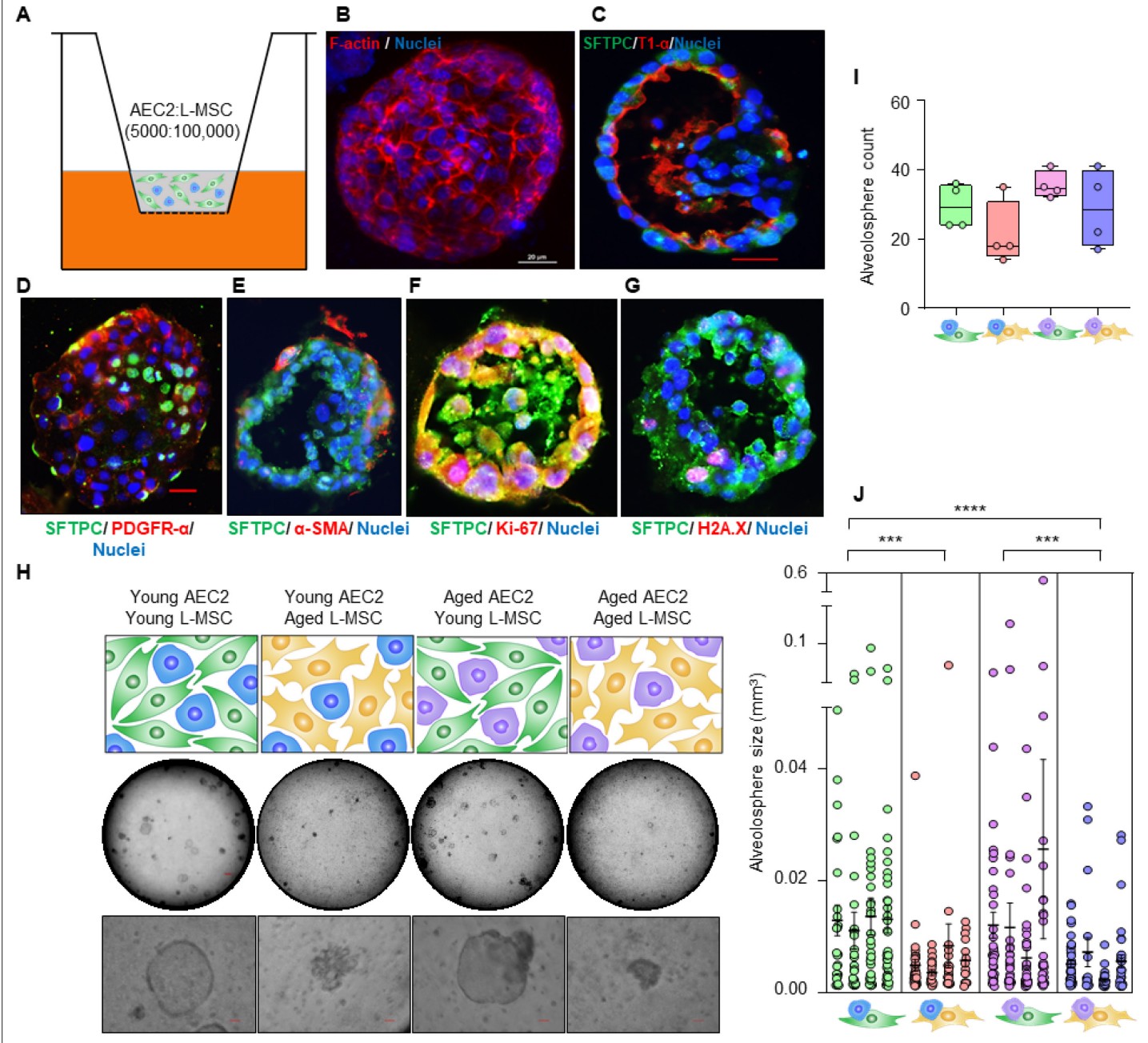

**Figure 1.** Aging lung-mesenchymal stromal cells (L-MSCs) impair self-organization of alveolar epithelial stem cells (AEC2s) and alveolosphere formation. (**A**) Alveolosphere assay. AEC2s and L-MSCs were purified from the young (3 months) mice lungs. AEC2s (5000 cells/well) and L-MSCs (100,000 cells/well) were mixed and seeded in Matrigel: MTEC plus media (1:1) and co-cultured in cell-culture inserts in 24-well dishes as shown. Alveolospheres form within 9–12 days of co-culture. (**B**) Alveolosphere whole mounts were immunofluorescently stained with antibody against F-actin (red) for confocal imaging. Image showing maximum intensity projection of an alveolosphere (scale bar = 20 μm). (**C**) Immunofluorescence (IF) staining showing localization of AEC2s (surfactant protein-C [SFTPC], green) and AEC1s (lung type I integral membrane glycoprotein-α [T1-α], red) within the alveolospheres. (**D**) IF staining showing localization of platelet-derived growth factor receptor-α (PDGFR-α, red) expressing alveolar L-MSCs within the alveolospheres. (**E**) Alpha-smooth muscle actin (α-SMA, red) IF staining showing presence of myofibroblasts in the alveolosphere. (**F, G**) Cell proliferation and DNA repair/apoptosis within the alveolospheres were determined by Ki-67 (**F**; red) and histone H2A.X (**G**; red) IF staining, respectively. Nuclei were stained with Hoechst 33,342 (blue; scale bars = 20 μm). (**H**) L-MSCs and AEC2s from young (3 months) and aged mice (24 months) were co-cultured in varied combinations as shown (upper panel); alveolospheres were imaged by brightfield microscopy. Images at low and high magnifications are shown (middle panel, scale bar = 300 μm; lower panel, scale bar = 20 μm). (**I**) Alveolospheres in each well were counted (n = 4 mice); box and whiskers plot showing median alveolosphere count in each group (p>0.05; ANOVA; Tukey's pairwise comparison test). (**J**) Alveolosphere sizes (volumes) were determined for each of the co-culture groups using ImageJ v1.47 software. Nested scatterplot showing mean ± SEM of all the alveolospheres counted in each well for

*Figure 1 continued on next page*

*Figure 1 continued*

each group (n = 4 mice; ***p<0.001, young AEC2s:young L-MSCs vs. young AEC2s:aged L-MSCs; aged AEC2s:young L-MSCs vs. aged AEC2s:aged L-MSCs; ****p<0.0001, young AEC2s:young L-MSCs vs. aged AEC2s:aged L-MSCs; ANOVA followed by Tukey's pairwise comparison test).

The online version of this article includes the following source data and figure supplement(s) for figure 1:

**Source data 1.** Alveolosphere formation with varying combinations of L-MSCs and AEC2s.

**Figure supplement 1.** Measurements of alveolosphere size and number.

cell proliferation marker, Ki-67, and the double-strand DNA damage repair marker, histone H2A.X, suggesting ongoing AEC2 turnover within these 3D organoids (*Figure 1F and G*). This self-organizing behavior of L-MSCs and AEC2s is critically dependent on the presence of both cell types as AEC2s in the absence of AEC2s do not self-organize to form alveolospheres (*Figure 1—figure supplement 1A*). Together, these data support the crosstalk between L-MSCs and AEC2s that permit formation of distinct alveolar organoid-like structures; this intercellular communication may be perturbed during aging accounting for age-associated loss of AEC2 regeneration/maintenance.

To determine the effect of age on cellular self-organization and alveolosphere formation, L-MSCs and AEC2s were isolated from the lungs of young (2–3 months) and aged (22–24 months) mice and seeded in various combinations (*Figure 1H*, upper panel). L-MSCs and AEC2s from young mice generated normal alveolospheres, whereas L-MSCs and AEC2s from the lungs of aged mice showed impaired self-organization with condensed, poorly formed organoids. Interestingly, combining aged AEC2s with young L-MSCs resulted in relatively normal-appearing alveolospheres, while the reverse combination (young AEC2s and old L-MSCs) did not (*Figure 1H*, middle and lower panels; *Figure 1—figure supplement 1B*). Quantitative analyses revealed that, while the number of alveolospheres formed was not significantly different between groups (*Figure 1I*), the size of alveolospheres was critically dependent on the age of L-MSCs, and not that of AEC2s (*Figure 1J*). These findings indicate that secreted factor(s) from young L-MSCs are capable of supporting the self-organizing behavior of alveolospheres, while aged L-MSCs do not.

Cellular senescence accumulates in tissues with advancing age (*Krishnamurthy et al., 2004*) and has been proposed as a key driver of aging and aging-related disease phenotypes (*Baker et al., 2016*; *Kennedy et al., 2014*). Based on our observation that aged L-MSCs were incapable of supporting alveolosphere formation, we explored whether the emergence of cellular senescence may account for this finding. First, we confirmed the senescent features of L-MSCs from aged mice in monolayer 2D cell culture (*Figure 2A*, *Figure 2—figure supplement 1*) and in 3D alveolospheres (*Figure 2B*) by staining for β-galactosidase (β-gal) (*Dimri et al., 1995*) and lipofuscin (*Georgakopoulou et al., 2013*), respectively. To characterize secreted factors that may account for the age-associated dysregulation of cell-cell communication, we first measured cytokines/growth factors secreted from both young and aged L-MSCs using antibody arrays (*Figure 2C*). A number of cytokines that have been associated with the senescence-associated secretory phenotype (SASP) were found to be released at higher levels by aged L-MSCs, including IL-6, CCL12/MCP-5, CCL11/Eotaxin, and WISP-1/CCN4; in contrast, IGFBP1 was the only secreted protein that was statistically more elevated in young L-MSCs in this cytokine array (*Figure 2D and E*).

In addition to predefined cytokine array analyses, we employed an unbiased approach to identification of secreted proteins from young and aged L-MSCs by mass spectrometry-based discovery proteomics (*Figure 2F*). After adjustments for false discovery rates, 503 high-confidence proteins were identified; of these, 235 proteins were found to have non-zero quantifiable values in at least three of four experimental repeats per group for subsequent statistical analysis. The 36 proteins that passed both a single pairwise statistical test (p<0.05) in addition to a fold change of ±1.5 (*Supplementary file 1A*) were subjected to principal component analysis (PCA) (*Figure 2G*) and heat map analysis (*Figure 2H*; quantitation of heat map proteins, *Figure 2I*). Gene ontology localization analyses revealed enrichment in secreted proteins, including extracellular vesicles and extracellular matrix (*Supplementary file 1B*); interestingly, gene ontology processes analysis indicated the top cellular processes as negative regulation of reactive oxygen species metabolic process and extracellular matrix/structure organization (*Supplementary file 1C*). Enrichment by top toxic pathologies revealed proteins associated with lung fibrosis (*Supplementary file 1D*), a disease associated with aging (*Thannickal et al., 2014*). Network analysis revealed upregulation of pathways associated with the canonical

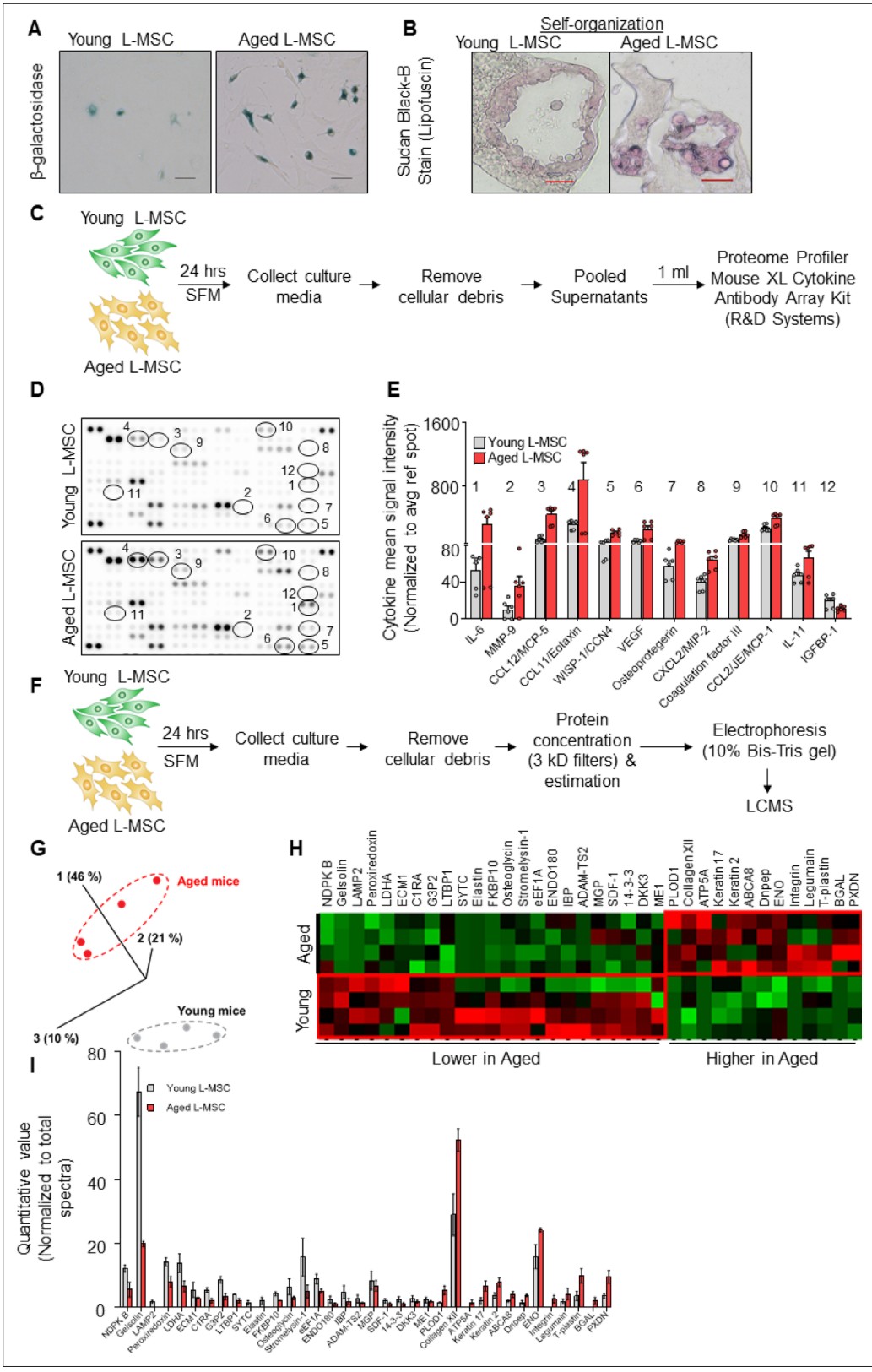

**Figure 2.** Aged lung mesenchymal stromal cells (L-MSCs) show redox imbalance and acquire senescence-associated secretory phenotype. (**A**) $\beta$-Galactosidase activity assay. Young and aged L-MSCs were plated in 6-well dishes at a density of 150,000 L-MSCs/well. L-MSCs were allowed to attach overnight and stained for $\beta$-galactosidase activity (scale bar = 50 μm). (**B**) Lipofuscin granules were detected in the alveolosphere paraffin

*Figure 2 continued on next page*

*Figure 2 continued*

sections by Sudan Black B staining. Sections were also stained with nuclear fast red for contrast (scale bar = 20 µm). (**C**) Cytokine array. 150,000 young and aged L-MSCs were plated in each well of a 6-well culture dish and grown overnight. Cells were washed with PBS and cultured for 24 hr in serum-free media (SFM; 1.5 ml/well). The culture media were pooled for each cell type and centrifuged to remove any cell and debris. 1 ml of supernatant was applied to antibody array, dotted with antibodies against 111 mouse cytokines and growth factors in duplicates. (**D**) Antibody array showing comparative expression of cytokines and growth factors in the culture media obtained from young and aged L-MSCs. Cytokines and growth factors showing significant difference are numbered and circled. (**E**) Signal intensity was determined for each dot using Image Quant array analysis software; mean signal intensity was calculated for each cytokine, and plotted. Twelve proteins with statistically significant difference (n = 6 samples from three mice; $p<0.05$; unpaired t-test) between the young and aged L-MSCs are shown. Data presented here include pooled data from three independent experiments. (**F**) Proteomics/mass spectrometry analysis. Cell culture media were collected from young and aged L-MSCs ($10^6$ cells) after 24 hr of growth in SFM in 10 cm dishes and subjected to proteomics analysis by liquid chromatography-mass spectrometry (LCMS). (**G**) A three-dimensional principal component analysis (PCA) plot showing replicated samples (young and aged) are relatively similar in their protein expression profiles and grouped together. (**H**) Heat map showing comparative expression of highly secreted proteins in the culture media between young and aged L-MSCs (n = 4 mice). (**I**) The top 36 proteins showing statistically significant difference (n = 4 mice; $p<0.05$; unpaired t-test) between the young and aged L-MSC secretome are plotted. Gene ontology, processes, and network analyses were performed on the mass spectrometry data, and results are provided in *Supplementary file 1*.

The online version of this article includes the following source data and figure supplement(s) for figure 2:

**Source data 1.** Cytokine array and mass spectrometry data comparing young and aged L-MSCs.

**Figure supplement 1.** Aged lung mesenchymal stromal cells (L-MSCs) show features of senescence.

WNT-signaling pathway and cell cycle regulation (*Supplementary file 1E*). Together, these findings suggest that aged L-MSCs are characterized by cellular senescence associated with SASP factors, oxidative stress, and alterations in regenerative pathways that may account for impaired alveolosphere formation.

Altered cellular metabolism has been linked to both senescence and aging (*López-Otín et al., 2016*; *Finkel, 2015*; *Sun et al., 2016*). To compare the energy state between young and aged lung L-MSCs, rates of mitochondrial respiration and glycolysis were analyzed by Seahorse XF Analyzer. Real-time oxygen consumption rates (OCR) and extracellular acidification rates (ECAR) were significantly higher in aged L-MSCs as compared to young L-MSCs (*Figure 3A and B*). Aged L-MSCs showed significantly higher basal, maximal, ATP-linked, proton leak, and non-mitochondrial respiration when compared to young L-MSCs with no change in reserve capacity (*Figure 3C-H*). Higher non-mitochondrial OCRs in aged L-MSCs suggest potential activation of oxygen-metabolizing NADPH oxidases in aged L-MSCs. An energy map profile of the OCR and ECAR data indicated a higher basal rate of both mitochondrial respiration and glycolysis (*Figure 3I*), suggesting that aged L-MSCs are under a basal state of higher metabolic demand. This higher energy demand under conditions of equivalent nutrient supply was associated with reduced levels of ATP/ADP under steady-state basal conditions (*Figure 3J*). Measurement of ECAR indicated that, despite similar basal ECAR, higher rates of maximal and non-glycolysis related ECAR were observed in aged L-MSCs. (*Figure 3—figure supplement 1*). Together, these data indicate that senescence of L-MSCs is characterized by higher baseline metabolic demand with reduced bioenergetic efficiency.

Hydrogen peroxide ($H_2O_2$) has emerged as critical regulator of redox signaling and oxidative stress (*Sies, 2017*). Replication-induced senescence of human lung fibroblasts results in higher rates of extracellular $H_2O_2$ release in association with increased expression of NADPH oxidase 4 (Nox4) (*Sanders et al., 2015*), a gene that is inducible by the pro-senescent/pro-fibrotic mediator, transforming growth factor-β1 (TGF-β1) (*Hecker et al., 2009*). We explored whether L-MSCs isolated from naturally aged mice are associated with higher Nox4-dependent $H_2O_2$ secretion; in comparison to young (3 months old) L-MSCs, aged (22–24 months old), L-MSCs released higher baseline levels of $H_2O_2$, an effect that was further stimulated by exogenous stimulation with TGF-β1 (*Figure 4A*). This age-dependent, pro-oxidant phenotype of L-MSCs was markedly reduced in aged mice heterozygous for Nox4 (*Nox4$^{-/+}$*; *Figure 4B and C*), implicating Nox4 as a critical mediator of both basal and TGF-β1-induced $H_2O_2$ release. Although it is difficult to replicate the tonic, continuous release of $H_2O_2$ from a constitutive enzymatic source such as Nox4 in metabolically active cells, we tested the effect of a bolus addition

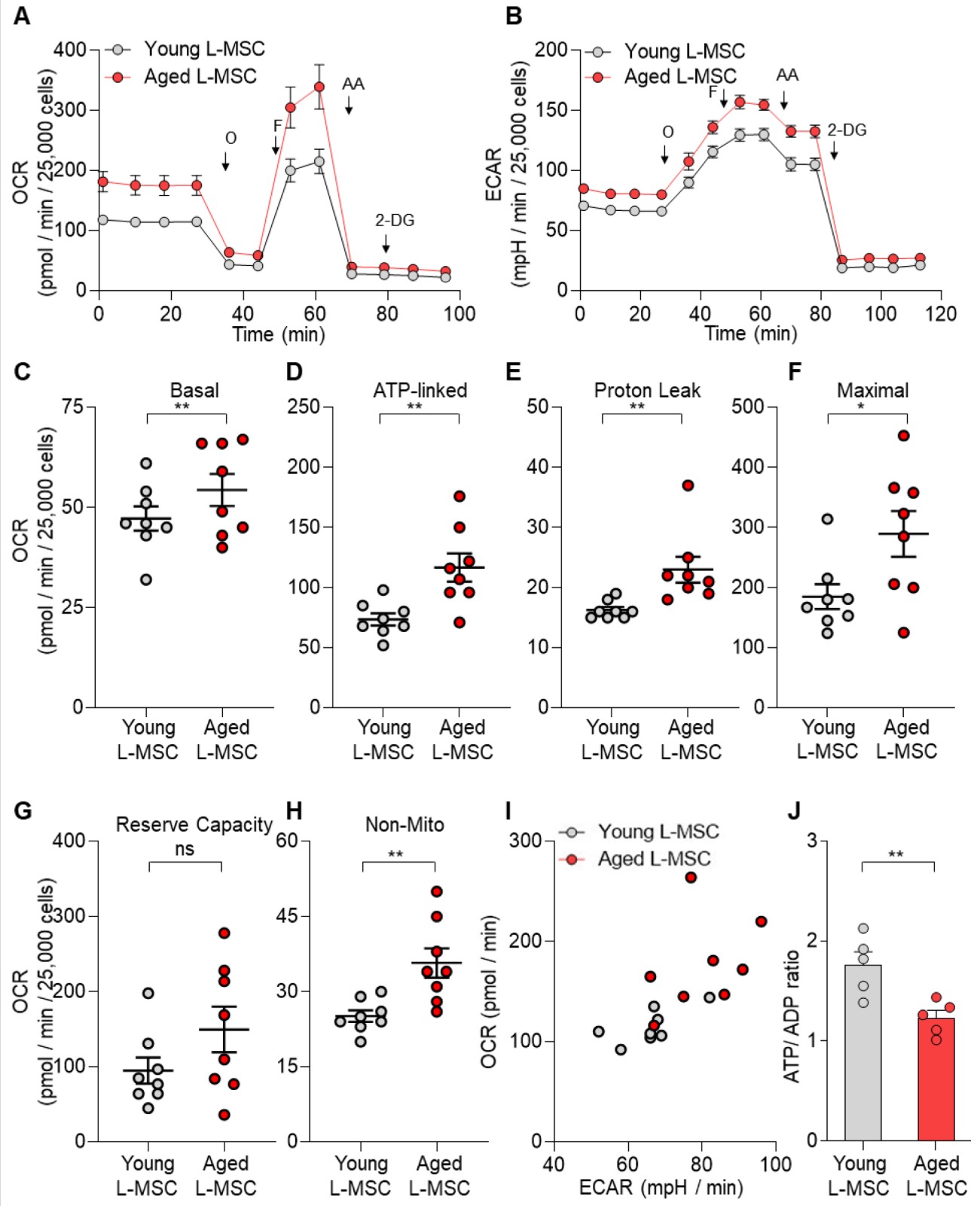

**Figure 3.** Senescent lung mesenchymal stromal cells (L-MSCs) demonstrate altered bioenergetics. Young and aged mouse L-MSCs were grown in complete Dulbecco's Modified Eagle Medium (DMEM) for 24 hr. 25,000 cells were seeded in Seahorse XF-24 cell culture microplates. The cells were treated sequentially with mitochondrial inhibitors: oligomycin (Oligo), carbonyl cyanide 4-(trifluoromethoxy) phenylhydrazone (FCCP), antimycin (AA), and glycolytic inhibitor: 2-deoxy-d-glucose (2-DG). (**A**) Real-time oxygen consumption rates (OCRs) and (**B**) real-time extracellular acidification rates

*Figure 3 continued*

(ECARs) between the young and aged L-MSCs were compared. (**C–H**) Basal, ATP-linked, proton leak, maximal, reserve capacity, and non-mitochondria-related OCRs were calculated and plotted (n = 8 mice; 10 technical replicates of each averaged; *p<0.05, **p<0.01; unpaired t-test; vs. young L-MSCs). (**I**) An energy map was generated from the OCR and ECAR data (above) showing higher basal rate of both mitochondrial respiration and glycolysis in aged L-MSCs. (**J**) ATP/ADP ratio. Young and aged L-MSCs (10,000 cells) were plated in 96-well flat-bottomed dish and allowed to attach. Cells were treated for 5 min with nucleotide-releasing buffer. Relative ATP and ADP levels were measured from luminescent conversion of ATP-dependent luciferin by the luciferase enzyme. ATP/ADP ratio was calculated and plotted. Graph showing comparative ATP/ADP ratio between young and aged L-MSCs (n = 5 technical replicates; **p<0.01; unpaired t-test; vs. young L-MSCs).

The online version of this article includes the following source data and figure supplement(s) for figure 3:

**Source data 1.** Bioenergetics of senescent L-MSCs.

**Figure supplement 1.** Aged lung mesenchymal stromal cells (L-MSCs) show features of senescence and altered bioenergetics.

**Figure supplement 1—source data 1.** ECAR measurements in young and aged L-MSCs.

of $H_2O_2$ (1 µM) to organoid culture comprising young AEC2s and young L-MSCs; this resulted in a decrease in the size of alveolospheres, without significantly affecting the numbers of alveolospheres formed (*Figure 4—figure supplement 1A–C*).

Recent studies implicate Nox4 in metabolic reprogramming (*Bernard et al., 2017*), although the effects of Nox4 on age-related metabolic dysfunction are unclear. The reduced levels of ATP/ADP in aged L-MSCs were partially rescued in age-matched $Nox4^{-/+}$ L- MSCs (*Figure 4D*), implicating a role for Nox4 in the reduced bioenergetic efficiency associated with aging. The steady-state levels of the tricarboxylic acid (TCA) cycle metabolites, malate and citrate, were reduced in wild-type aged L-MSCs; these levels recovered to levels similar to young L-MSCs in $Nox4^{-/+}$ L- MSCs, although only malate achieved statistical significance (*Figure 4E and F*). Levels of other TCA metabolites that were not significantly altered with aging in L-MSCs were not affected by a deficiency in Nox4 (*Figure 4— figure supplement 1D-G*). β-gal activity was found to be significantly lower in Nox4-deficient aged L-MSCs as compared to wild-type aged L-MSCs (*Figure 4—figure supplement 1H*). In addition to metabolic dysfunction, an important feature of senescence reprogramming is the activation of a SASP program. We determined whether Nox4 modulates the release of cytokines/growth factors by aged L-MSCs. Protein (antibody-based) arrays showed significant downregulation of SASP-associated cytokines in aged $Nox4^{-/+}$ L -MSCs vs. aged L-MSCs (*Figure 4—figure supplement 1I*). We were unable to identify a group of cytokines (except for coagulation factor III) that were specifically identified in the dataset comparing young vs. aged L-MSCs based on the antibody-based cytokine array method. Further studies with an unbiased approach such as that employed by mass spectrometric analyses (*Figure 2I*) will be required for more in-depth analyses of Nox4 regulated SASP cytokines/growth factors. Interestingly, Nox4 haploinsufficiency in aged L-MSCs resulted in significant downregulation of OCR as compared to aged L-MSCs (*Figure 4G*). $Nox4^{-/+}$ aged L-MSCs showed significantly lower basal, ATP-linked, proton leak, and maximal respiration compared to wild-type aged L-MSCs. Non-mitochondrial respiration was also lower in aged $Nox4^{-/+}$ L- MSCs, although this did not achieve statistical significance (*Figure 4H*). Together, these studies indicate that Nox4 contributes to metabolic dysfunction, bioenergetic inefficiency, and SASP programming in L-MSC aging.

Based on the observations that Nox4 alters cellular bioenergetics of L-MSCs, we examined the effects of Nox4 on regulating the self-organizing potential of alveolospheres. The aberrant formation of alveolospheres when young AEC2s were cultured with aged L-MSCs were rescued when replaced with age-matched $Nox4^{-/+}$ L- MSCs (*Figure 4I*; *Figure 4—figure supplement 1J*); quantitation of both alveolospheres counts (*Figure 4J*) and alveolosphere size (*Figure 4K*) was significantly enhanced when these assays were conducted with Nox4-deficient L-MSCs. We were also unable to detect significant changes in alveolosphere formation when the Nox4 inhibitor, GKT137831 (10 µM), was added to an organoid culture system of young AEC2s and aged L-MSCs (*Figure 4—figure supplement 1K-M*); however, it is difficult to ascertain whether this intervention effectively inhibited Nox4 activity in this organoid model. While the phenotype of impaired alveolosphere formation in young AECs and aged L-MSCs was reversed by haploinsufficiency of Nox4 in L-MSCs (*Figure 4I–K*), this was not sufficient to completely reverse the phenotype in aged AEC2s (*Figure 4—figure supplement 1N-P*). This finding, along with the observation that wild-type young L-MSCs are sufficient to reverse the aged AEC2 phenotype (*Figure 1H*), suggests that in addition to reduced expression of Nox4 in aged L-MSCs, the young mesenchyme may support niche function and recovery of alveolosphere formation of aged

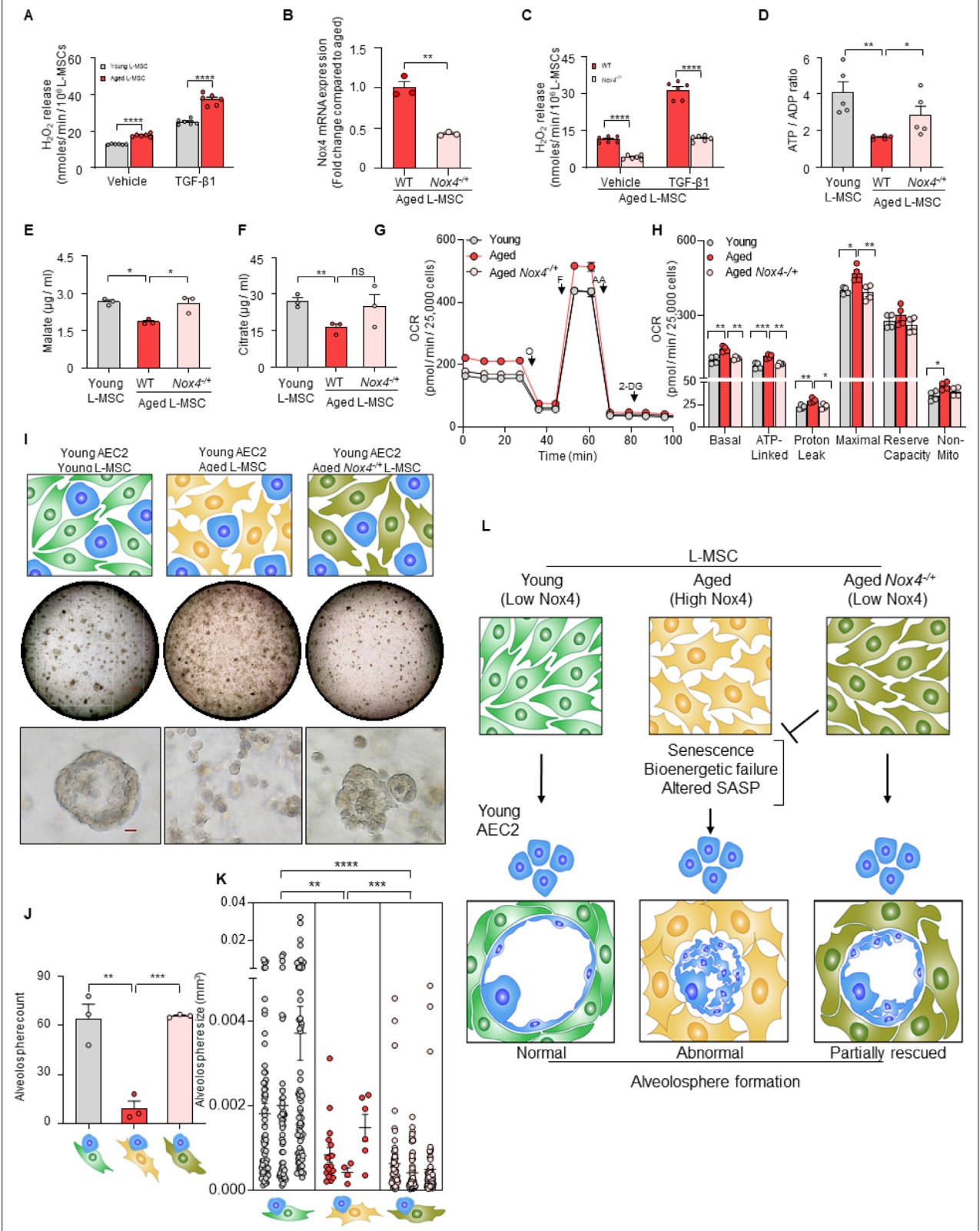

**Figure 4.** Nox4 deficiency in aged lung mesenchymal stromal cell (L-MSC) improves bioenergetics and restores type 2 alveolar epithelial cell (AEC2) self-organization. (**A**) Hydrogen peroxide ($H_2O_2$)-release assay. Young and aged L-MSCs were grown in very low serum (1%) containing media for 24 hr and treated with either transforming growth factor-$\beta$1 (TGF-$\beta$1) or vehicle for 16 hr, and real-time $H_2O_2$ release was determined. Here, bar graph showing comparative $H_2O_2$ release between young and aged L-MSCs with or without TGF-$\beta$1 treatment (n = 6 technical replicates; ****p<0.0001;

*Figure 4 continued on next page*

*Figure 4 continued*

unpaired t-test). (**B**) RT-PCR analysis. Total RNA was isolated from aged and aged $Nox4^{-/+}$ L -MSCs, and subjected to real-time PCR analysis for Nox4 mRNA expression. Data were normalized to $\beta$ -actin and represented graphically as fold change compared to aged L-MSCs (n = 3 mice; three technical replicates of each averaged; **p<0.01; unpaired t-test). (**C**) Data showing comparative $H_2O_2$ release between aged and aged $Nox4^{-/+}$ L- MSCs at baseline and after TGF- $\beta$ 1 treatment (n = 6 technical replicates; ****p<0.0001; unpaired t-test). (**D**) Bar graph showing comparative ATP/ADP ratio between young, aged, and aged $Nox4^{-/+}$ L -MSCs (n = 5 technical replicates; **p<0.01, *p<0.05). (**E, F**) Targeted metabolomics. Relative levels of tricarboxylic acid cycle metabolites were determined and compared between young, aged, and aged $Nox4^{-/+}$ L -MSCs. Concentrations of malate (**E**) and citrate (**F**) are shown here (n = 3 technical replicates; *p<0.05; **p<0.01). (**G, H**) Real-time oxygen consumption rates (OCRs) between the young, aged, and aged $Nox4^{-/+}$ L -MSCs were compared. Basal, ATP-linked, proton leak, maximal, reserve capacity, and non-mitochondria-related OCRs were calculated and plotted (n = 4 mice/group; 10 technical replicates of each averaged; *p<0.05, **p<0.01; ***p<0.001). (**I**) Alveolosphere assay. Young, aged, and aged $Nox4^{-/+}$ L -MSCs were co-cultured with young AEC2s in a ratio described earlier (upper panel). The alveolospheres were imaged by brightfield microscopy after 12 days of co-culture, and comparative outcomes are shown here in low (middle panel; scale bar = 300 µm) and higher magnifications (lower panel; scale bar = 20 µm). (**J**) Alveolospheres in each well were counted (mean ± SEM; n = 3 mice; **p<0.01; ***p<0.001). (**K**) Alveolosphere sizes (volumes) were determined for each of the three co-culture groups. Nested scatterplot showing mean ± SEM of all the alveolospheres counted in each well for each group (n = 3 mice; **p<0.01, ***p<0.001, ****p<0.0001). (**D–K**) Statistical analysis: ANOVA followed by Tukey's pairwise comparison test. (**L**) Schematic summarizing the important findings from this study.

The online version of this article includes the following source data and figure supplement(s) for figure 4:

**Source data 1.** Functional characteristics of wild type and Nox4-deficient aged L-MSCs.

**Figure supplement 1.** Role of Nox4 in senescent L-MSCs and alveolosphere formation.

**Figure supplement 1—source data 1.** Characterization of metabolism, SASP profile, and alveolosphere formation in Nox4-deficient L-MSCs.

AEC2s through Nox4-independent mechanisms. Thus, an aging alveolar stem cell niche may involve both a gain of 'pro-senescent factors' (Nox4) and loss of 'rejuvenation factors' by the mesenchyme; the identification of the latter will require further study. Together, our studies support a critical role for Nox4-dependent oxidative stress, senescence, and bioenergetic insufficiency in restricting the capacity for cellular self-organization in a 3D organoid model of aging and stem/progenitor cell function (*Figure 4L*).

A fundamental difference between non-living matter and life forms on our planet is the ability to extract energy from the environment, primarily through oxidation of carbon-based fuels. Thus, the second law of thermodynamics that governs both living and non-living matter does not strictly apply to living organisms capable of self-renewal through the continuous and dynamic exchange of energy (from exogenous nutrients) and mass (biosynthetic and degradative processes in respiring tissues/organs). It follows then that, when this capacity for energy-dependent self-renewal is diminished, the law of entropic degeneration will be operative in living organisms. Aging is associated with derangements in metabolism that influences, and may in fact control, many of the well-recognized hallmarks of aging, including cellular senescence (*López-Otín et al., 2016*; *López-Otín et al., 2013*). The inter-relatedness of metabolism with these traditional aging hallmarks is difficult to untangle due to complexities of studies in living organisms and the relative simplicity of 2D cell culture models.

Organoids offer the opportunity to study complex living phenomena such as emergent properties and self-organization in biological systems while at the same time allowing for reductionist approaches that provide mechanistic understanding of these complex processes. Using a combination of methods that included a 3D organoid model, biochemical approaches, proteomics, and gene targeting, we demonstrate a critical role for the oxygen metabolizing enzyme, Nox4, in regulating cellular bioenergetics and senescence that limits stem cell function. Consistent with contemporary theories on aging, Nox4 may function as an antagonistically pleiotropic gene and contribute a number of age-related degenerative disorders, including fibrotic diseases (*Thannickal, 2010*; *Lambeth, 2007*). Interestingly, proteomic analyses of proteins secreted by aged L-MSCs in the current study suggested alterations in cellular redox/oxidative stress (from gene ontology analysis) and lung fibrosis (as a 'toxic pathology'). These findings, as well as candidate SASP factors, highlight the emerging importance of intercellular communication and stem cell exhaustion as important hallmarks of aging (*López-Otín et al., 2013*).

Regenerative mechanisms in adult, mammalian organisms primarily rely on the activation and differentiation of tissue/organ-resident stem cells (*Hogan et al., 2014*). Homeostatic maintenance of these stem cells requires a tightly regulated niche that includes other supporting cell types, in particular stromal cells (*Basil et al., 2020*). An important finding from our studies is the observation that L-MSC aging, relative to AEC aging, largely accounts for the age-related inability to form

alveolospheres. Thus, aging of the stem cell niche may be as important, and perhaps more decisive, in promoting certain age-related pathologies. Therapeutic targeting of metabolic aging within the stem cell niche, specifically that of lung-resident fibroblasts/myofibroblasts, may offer a more effective and viable strategy for age-related degenerative disorders such as tissue/organ fibrosis.

## Materials and methods

### Key resources

All reagents, antibodies, assay kits, and mice used in this study are listed in Appendix 1—key resources table.

All animal protocols were approved by the Institutional Animal Care and Use Committees (IACUC) at the University of Alabama at Birmingham. The mice were acclimatized in the animal facility at least for a week before experiments. Male mice were used in this study for their greater susceptibility to age-related diseases.

### Cell culture

Mouse lung L-MSCs were isolated and propagated following protocol developed in our laboratory (*Vittal et al., 2005*). L-MSCs were isolated by collagenase digestion of the lungs and anchorage-dependent ex vivo growth on plastic culture dishes in Dulbecco's Modified Eagle Medium (DMEM) supplemented with 10 % fetal bovine serum, 4 mM L-glutamine, 4.5 g/L glucose, 100 U/ml penicillin, 100 µg/ml streptomycin, and 1.25 µg/ml amphotericin B (Fungizone), in a humified chamber at 37 °C in 5% $CO_2$, 95 % air. AEC2s were isolated and purified from the young and aged mice lungs via enzymatic digestion (Dispase II), cell-specific antibody labeling (biotinylated Ter-119, CD104, CD16/32, CD45, CD31), and magnetic separation (Anti-Biotin MicroBeads) following published protocols (*Sinha and Lowell, 2016*) and used immediately. AEC2 purity was determined by fluorescent detection of EpCam$^+$/SFTPC$^+$/CD45$^-$/CD31$^-$ cell population using flow cytometry (*Sinha and Lowell, 2016*; *Bertoncello and Mcqualter, 2011*). L-MSCs and AEC2s were seeded in Matrigel mixed with MTEC/plus media (*You et al., 2002*) (1:1) in cell culture inserts (24-well format, 0.4 µm pore size), and co-cultured in MTEC/Plus media as shown in *Figure 1A*.

### Immunofluorescence staining

Matrigel containing alveolospheres were fixed in 4 % paraformaldehyde, embedded in HistoGel, dehydrated, and paraffin embedded using standard protocol. 5-µm-thick sections were cut and mounted on glass slides and immunostained for mouse SFTPC, T1-α, PDGFRα, α-SMA, Ki-67, and histone 2A.X (H2A.X). Briefly, alveolosphere sections were deparaffinized in xylene and hydrated through ethanol series and water. Antigen retrieval was performed using citrate buffer at pH 6.0 in a 95 °C water bath. Tissue sections were blocked using 5 % normal goat or donkey serum and were then incubated in primary antibodies overnight at 4 °C. Appropriate IgG isotype controls were also used to determine specificity of staining. Secondary detection was performed using anti-mouse Alexa Fluor 594/488-tagged secondary antibodies. Nuclei were counterstained with Hoechst 33342 dye for immunofluorescence detection. The stained alveolosphere sections were mounted in Vectashield and viewed and imaged in a Keyence BZ-X710 inverted microscope with brightfield as well as fluorescent imaging capability.

### Histochemical staining

β-Galactosidase staining was performed on young and aged L-MSCs in culture, according to instructions provided in the Senescence Detection Kit (Abcam). Lipofuscin staining was performed on paraffin sections of alveolospheres following published protocol (*Georgakopoulou et al., 2013*).

### Cytokine array

Cytokine array was carried out in the young and aged L-MSC culture media using Proteome Profiler Mouse XL Cytokine Array kit (R&D Systems) as per the manufacturer's instructions.

### Proteomics/mass spectrometry analysis

Proteomics analysis was carried out following established protocols (*Ludwig et al., 2016*) with minor changes. Briefly, 5 ml of each cell culture media specimen were concentrated using Amicon Ultra

4 ml, 3 kDa molecular weight cutoff filters (Millipore) and protein concentrations were determined using Pierce BCA Protein Assay Kit. Proteins (10 µg) per sample were reduced with DTT and denatured at 70 °C for 10 min prior to loading onto Novex NuPAGE 10 % Bis-Tris Protein gels (Invitrogen, Cat. # NP0315BOX). The gels were stained overnight with Novex Colloidal Blue Staining kit (Invitrogen, Cat. # LC6025). Following destaining, each lane was cut into 3 MW fractions and equilibrated in 100 mM ammonium bicarbonate (AmBc), each gel plug was then digested overnight with Trypsin Gold, Mass Spectrometry Grade (Promega, Cat. # V5280) following the manufacturer's instruction. Peptide digests (8 µl each) were injected onto a 1260 Infinity nHPLC stack (Agilent Technologies) and separated using a 75 µm I.D. × 15 cm pulled tip C-18 column (Jupiter C-18 300 Å, 5 µm, Phenomenex). This system runs in-line with a Thermo Orbitrap Velos Pro hybrid mass spectrometer, equipped with a nano-electrospray source (Thermo Fisher Scientific), and all data were collected in CID mode. Searches were performed with a species-specific subset of the UniprotKB database. The list of peptide IDs generated based on SEQUEST (Thermo Fisher Scientific) search results was filtered using Scaffold (Protein Sciences, Portland, OR). Gene ontology assignments and pathway analysis were carried out using MetaCore (GeneGO Inc, St. Joseph, MI). The original dataset has been submitted to Dryad repository (doi:10.5061/dryad.x0k6djhj1).

## Extracellular flux analysis

Analyses of cellular bioenergetics were performed using the Seahorse XFe96 Extracellular Flux Analyzer (Agilent, Santa Clara, CA). L-MSCs from young and aged mice were cultured for 24 hr in low serum (1%) containing media. Cells were then detached and replated at a density of 25,000 cells/well into a XF96 microplate 5 hr prior to assay in the same media. After the cells were attached, XF-DMEM media (DMEM supplemented with 5.5 mM glucose, 1 mM pyruvate, and 4 mM L-glutamine, pH 7.4 at 37 °C) was added to the wells, and cells were incubated for 1 hr prior to assay in a non-$CO_2$ incubator at 37 °C. Mitochondrial stress test, a parallel measure of basal OCR and ECAR, was carried out following sequential injections of oligomycin (1 µg/ml), carbonyl cyanide 4-(trifluoromethoxy) phenylhydrazone (FCCP [0.6 µM]), antimycin-A (10 µM), and 2-deoxyglucose (50 µM).

## ATP/ADP ratio

This assay was performed using an ADP/ATP ratio assay kit (Abcam) following the manufacturer's instructions.

## $H_2O_2$-release assay

This assay was performed based on protocol developed in our laboratory for quantitative measurement of extracellular $H_2O_2$ release from adherent fibroblasts (*Thannickal and Fanburg, 1995*). This fluorimetric method relies on the oxidative conversion of homovanillic acid (HVA), a substituted phenolic compound, to its fluorescent dimer in the presence of $H_2O_2$ and horseradish peroxidase (HRP). For this experiment, 150,000 young, aged, and aged *Nox4*[-/+] mouse lung L-MSCs were seeded in 6-well culture dishes. Next day, cells were serum-starved (1%) overnight and treated with either vehicle or 2.5 ng/ml TGF-β1. Media was aspirated after 16 hr post-treatment; cells were washed with 1 ml of HBSS (without calcium and magnesium) and incubated for 2 hr in 1 ml of assay media containing 100 µM HVA and 5 U/ml HRP in HBSS (containing calcium and magnesium). Reactions were stopped adding stop solution and fluorescence was measured in a BioTek microplate reader at excitation and emission maximum of 321 nm and 421 nm, respectively. Extracellular $H_2O_2$ release was measured from standard curve generated from a known concentration of $H_2O_2$. The data are normalized to cell number, and $H_2O_2$ concentrations are presented as nanomoles/min/$10^6$ L-MSCs.

## Quantitative RT-PCR

Total RNA was isolated from aged and aged *Nox4*[-/+] L- MSCs using RNeasy Mini Kit (Qiagen) and reverse transcribed using iScript Reverse Transcription SuperMix for RT- qPCR (Bio-Rad). Real-timePCR reactions were performed using SYBR Green PCR Master Mix (Life Technologies) and gene-specific primer pairs for Nox4 and β-actin (Integrated DNA Technologies; for primer sequences, see Appendix 1—key resources table). Reactions were carried out for 40 cycles (95 °C for 15 s, 60 °C for 1 min) in a StepOnePlus Real Time PCR System (Life Technologies). Real-time PCR data ($2^{-\Delta\Delta Ct}$) is presented as Nox4 mRNA expression normalized to β-actin.

## Targeted metabolomics

Concentrations of TCA metabolites in the young and aged L-MSCs were determined by liquid chromatography tandem mass spectrometry (LCMS) following published protocols (*Tan et al., 2014*) with minor modifications. Dry stocks of analytically pure standards were weighed out and solubilized in MilliQ water and further diluted with 50 % methanol to a 10 µg/ml stock. This stock was diluted to generate concentrations for standards ranging from 0.5 to 1000 ng/ml. Methanolic sample extracts were dried under nitrogen ($N_2$) gas and reconstituted in 100 µl of 50 % methanol. Methanolic samples and standards were derivatized by addition of 10 µl 0.1 M O-benzylhydroxylamine hydrochloride (O-BHA) and 10 µl 0.25 M N-(3-dimethylaminopropyl)-N'-ethylcarbodiimide hydrochloride (EDC) at room temperature (RT) for 1 hr. Samples were then transferred to a 13 × 100 mm borosilicate tube for liquid-liquid extraction. Samples were then dried under $N_2$ gas at RT and were reconstituted in 100 µl of 0.1 % formic acid (FA) and transferred to HPLC vials for LCMS analysis. LCMS was carried out with a Prominence HPLC and API 4000 MS. 10 µl of sample was injected onto an Accucore 2.6 µm C18 100 × 2.1 mm column at 40 °C for gradient separation. Column eluent was directed to the API 4000 MS operating system, controlled by Analyst 1.6.2 software. Post-acquisition data analysis was carried out using MultiQuant v3.0.3 software. All standard curve regressions were linear with $1/x^2$ weighting.

## Statistical analysis

Sample size calculation was deemed unnecessary as our study contains no clinical component or in vivo intervention in mice and was therefore not computed. In this study, three cohorts of mice (young, aged, and aged-*Nox4*$^{-/+}$), consisting of 3–8 mice per cohort, were used to obtain sufficient quantities of AEC2s and L-MSCs to perform the necessary assays in biological and technical replicates. In this instance, each mouse in a given cohort was considered a biological replicate, while repeated measurements in cells obtained from each animal were considered technical replicates. Student's t-test was used to compare between two experimental groups. When more than two experimental groups were present, data were analyzed by multifactor analysis of variance (ANOVA). Pairwise comparisons were performed to identify which paired groups presented significant differences and were adjusted by Tukey's correction. Data were tested for normality distribution by Shapiro–Wilk test before comparative analysis. Non-normal data (alveolosphere size; *Figures 1J and 4K*) were natural log transformed and compared by ANOVA and pairwise comparisons. Differences between the experimental groups were considered significant when $p < 0.05$. All experiments were repeated 2–4 times for reproducibility. Measurement of TCA metabolites was performed only once due to unavailability of age-matched aged *Nox4*$^{-/+}$ mice in our colony.

## Acknowledgements

We thank Dr. Tingting Yuan for assistance with preparing paraffin blocks and cutting histological sections. We thank Dr. Robert Grabski and Mr. Shawn Williams for their assistance with confocal imaging at the High Resolution Imaging Facility at the University of Alabama at Birmingham (UAB). We also thank Dr. Stephen Barnes, Dr. Landon Wilson, and Mr. Taylor Berryhill at the Targeted Proteomics & Metabolomics Lab, UAB, for their assistance with measuring TCA cycle metabolites. This work was supported by NIH grants P01 HL114470, R01 AG046210, and R01 HL152246; and by VA Merit Award I01BX003056 (to VJT). DC is supported by NIH/NHLBI grant R01 HL139617 (to JZ and VJT); JAM is supported by NIH/NCI grant P30 CA013148; JSD is supported by NIH grants R01 HL128502 and P42 ES027723.

## Additional information

### Funding

| Funder | Grant reference number | Author |
| --- | --- | --- |
| NIH Office of the Director | R01 HL152246 | Victor J Thannickal |
| NIH Office of the Director | P01 HL114470 | Victor J Thannickal |

| Funder | Grant reference number | Author |
| --- | --- | --- |
| Veteran Affairs Administration | I01BX003056 | Victor J Thannickal |
| NIH Office of the Director | R01 AG046210 | Victor J Thannickal |
| NIH Office of the Director | R01 HL139617 | Jaroslaw W Zmijewski Victor J Thannickal |
| NIH Office of the Director | P30 CA013148 | James A Mobley |
| NIH Office of the Director | R01 HL128502 | Jessy S Deshane |
| NIH Office of the Director | P42 ES027723 | Jessy S Deshane |

The funders had no role in study design, data collection and interpretation, or the decision to submit the work for publication.

## Author contributions

Diptiman Chanda, Conceptualization, Data curation, Formal analysis, Investigation, Methodology, Project administration, Resources, Software, Supervision, Validation, Visualization, Writing – original draft, Writing – review and editing; Mohammad Rehan, Investigation, Software, Writing – review and editing; Samuel R Smith, Data curation, Formal analysis, Software, Writing – review and editing; Kevin G Dsouza, Investigation, Writing – review and editing; Yong Wang, Investigation; Karen Bernard, Methodology, Writing – review and editing; Deepali Kurundkar, Methodology, Resources; Vinayak Memula, Formal analysis, Software; Kyoko Kojima, Formal analysis, Investigation, Methodology, Resources; James A Mobley, Formal analysis, Methodology, Resources, Software, Writing – review and editing; Gloria A Benavides, Formal analysis, Methodology, Resources, Software; Victor Darley-Usmar, Data curation, Formal analysis, Resources, Supervision, Writing – review and editing; Young-iL Kim, Formal analysis, Resources, Writing – review and editing; Jaroslaw W Zmijewski, Stijn De Langhe, Writing – review and editing; Jessy S Deshane, Resources, Writing – review and editing; Victor J Thannickal, Conceptualization, Funding acquisition, Project administration, Resources, Supervision, Visualization, Writing – original draft, Writing – review and editing

## Author ORCIDs

Diptiman Chanda http://orcid.org/0000-0002-4835-4460
Stijn De Langhe http://orcid.org/0000-0003-3867-4572
Victor J Thannickal http://orcid.org/0000-0003-4266-8677

## Ethics

This study was performed in strict accordance with the recommendations in the Guide for the Care and Use of Laboratory Animals of the National Institutes of Health. All of the animals were handled according to approved institutional animal care and use committee (IACUC) protocols (#10105) of the University of Alabama at Birmingham. Mice were euthanized under isoflurane anesthesia before isolation of cells. Every effort was made to minimize suffering.

## Decision letter and Author response

Decision letter https://doi.org/10.7554/eLife.68049.sa1
Author response https://doi.org/10.7554/eLife.68049.sa2

# Additional files

## Supplementary files

• Supplementary file 1. Gene Ontology localizations, processes, and network analyses of mass spectrometry data.

• Transparent reporting form

## Data availability

Datasets associated with this article have been made available on Dryad (https://doi.org/10.5061/dryad.x0k6djhj1).

The following dataset was generated:

| Author(s) | Year | Dataset title | Dataset URL | Database and Identifier |
|---|---|---|---|---|
| Chanda D | 2021 | Data from: Discovery proteomics by mass spectrometry comparing secreted proteins from young and old mouse lung mesenchymal stromal cells | http://dx.doi:10.5061/dryad.x0k6djhj1 | Dryad Digital Repository, 10.5061/dryad.x0k6djhj1 |

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

## Appendix 1

**Appendix 1—key resources table**

| Reagent type (species) or resource | Designation | Source or reference | Identifiers | Additional information |
|---|---|---|---|---|
| Antibody | Podoplanin (T1-$\alpha$) monoclonal PE-Cyanine7 | eBioscience/Thermo Fisher Scientific | RRID:AB_2573459 | Host: Syrian hamster 0.125 µg/100 µl |
| Antibody | CD31 (PECAM-1) FITC | eBioscience/Thermo Fisher Scientific | RRID:AB_465012 | Host: rat 1 µg/100 µl |
| Antibody | CD326 (EpCAM) monoclonal APC-eFluor780 | eBioscience/Thermo Fisher Scientific | RRID:AB_2573986 | Host: rat 0.125 µg/100 µl |
| Antibody | CD24 monoclonal, APC | eBioscience/Thermo Fisher Scientific | RRID:AB_10870773 | Host: rat 0.06 µg/100 µl |
| Antibody | CD45 monoclonal (30-F11), PE | eBioscience/Thermo fisher Scientific | RRID:AB_465668 | Host: rat 0.125 µg/100 µl |
| Antibody | TER-119 monoclonal, biotin | eBioscience/Thermo fisher Scientific | RRID:AB_466796 | Host: rat 1:100 |
| Antibody | CD16/32, clone 2.4G2, biotin | BD Biosciences | RRID:AB_394658 | Host: rat 1:100 |
| Antibody | CD104 [346-11A] biotin | BioLegend | RRID:AB_961034 | Host: rat 1:100 |
| Antibody | CD31 [MEC13.3] biotin | BioLegend | RRID:AB_312910 | Host: rat 1:100 |
| Antibody | CD45, biotin | eBioscience/Thermo Fisher Scientific | RRID:AB_466446 | Host: rat 1:100 |
| Antibody | Ki-67/MKI67 polyclonal | Novus Biologicals | RRID:AB_10001977 | Host: rabbit 1:100 |
| Antibody | Podoplanin polyclonal | R&D Systems | RRID:AB_2268062 | Host: goat 15 µg/ml |
| Antibody | Phospho-histone H2A.X (Ser139) | Cell Signaling Technology | RRID:AB_2118010 | Host: mouse 1:100 |
| Antibody | Pro-surfactant protein C (SFTP-C) | Millipore Sigma | RRID:AB_91588 | Host: rabbit 1:1000 |
| Antibody | Platelet-derived growth factor receptor-$\alpha$ | R&D Systems | RRID:AB_2236897 | Host: mouse 1:50 |
| Antibody | $\alpha$-smooth muscle actin | American Research Product | RRID:AB_1540376 | Host: mouse 1:500 |
| Antibody | Goat anti-rabbit IgG secondary antibody-Alexa Fluor 594 | Thermo Fisher Scientific | RRID:AB_2556545 | Host: goat 2 µg/ml |
| Antibody | Donkey anti-goat IgG secondary antibody-Alexa Fluor 594 | Thermo Fisher Scientific | RRID:AB_2534105 | Host: donkey 2 µg/ml |
| Antibody | Donkey anti-mouse IgG secondary antibody-Alexa Fluor 594 | Thermo Fisher Scientific | RRID:AB_2556543 | Host: donkey 2 µg/ml |
| Antibody | Donkey anti-rabbit IgG secondary antibody-Alexa Fluor 488 | Thermo Fisher Scientific | RRID:AB_2556546 | Host: donkey 2 µg/ml |
| Antibody | Anti-Biotin MicroBeads UltraPure | Miltenyi Biotec | AB_2811216 | Host: mouse |

*Appendix 1 Continued on next page*

*Appendix 1 Continued*

| Reagent type (species) or resource | Designation | Source or reference | Identifiers | Additional information |
|---|---|---|---|---|
| Chemical compound, drug | Fluorescent cell staining reagents | Fixable Viability Dye eFluor 450 | eBioscience/Thermo Fisher Scientific | |
| Chemical compound, drug | Fluorescent cell staining reagents | Phalloidin-iFluor 594 Reagent -CytoPainter | Abcam | |
| Chemical compound, drug | Fluorescent cell staining reagents | Hoechst 33342 trihydrochloride trihydrate | Thermo Fisher Scientific | |
| Chemical compound, drug | Bovine pituitary extract | Millipore Sigma | P1476-2.5ML | MTEC/plus supplement |
| Chemical compound, drug | Collagenase, type 4 | Worthington Biochemical | LS004188 | L-MSC isolation |
| Chemical compound, drug | Dispase II | Millipore Sigma | 4942078001 | AEC2 isolation |
| Chemical compound, drug | Matrigel Membrane Matrix GFR | Corning | 356231 | Alveolosphere assay |
| Chemical compound, drug | Insulin-transferrin-selenium-ethanolamine (ITS -X) (100 X) | Thermo Fisher Scientific | 51500056 | MTEC/plus supplement |
| Chemical compound, drug | Human insulin soln. | Millipore Sigma | I9278-5ML | MTEC/plus supplement |
| Chemical compound, drug | EGF recombinant mouse protein | Thermo Fisher Scientific | PMG8041 | MTEC/plus supplement |
| Chemical compound, drug | STEMRD Y27632 ROCK inhibitor | Thermo Fisher Scientific | 50-175-997 | MTEC/plus supplement |
| Chemical compound, drug | All-trans retinoic acid | Millipore Sigma | R2625-100MG | MTEC/plus supplement |
| Chemical compound, drug | Cholera toxin | Millipore Sigma | C8052-5MG | MTEC/plus supplement |
| Chemical compound, drug | Recombinant human TGF-beta1 | PeproTech | 100-21-2ug | |
| Chemical compound, drug | Horseradish peroxidase | Millipore Sigma | P8375-25KU | |
| Chemical compound, drug | Homovanillic acid, fluorimetric reagent | Millipore Sigma | H1252-100MG | |
| Chemical compound, drug | Hydrogen peroxide solution, 30 % (w/w) | Millipore Sigma | H1009-100ML | |
| Chemical compound, drug | Oligomycin | Millipore Sigma | O4876-5MG | |
| Chemical compound, drug | Carbonyl cyanide 4 | Millipore Sigma | C2920-10MG | |
| Chemical compound, drug | Antimycin A | Millipore Sigma | A8674-25MG | |
| Chemical compound, drug | 2-deoxy D-glucose | Millipore Sigma | D6134-5G | |
| Chemical compound, drug | Vectashield | Vector Laboratories | H-1400-10 | |
| Chemical compound, drug | HistoGel; specimen processing gel | Richard-Allan Scientific/ Thermo Fisher Scientific | HG-4000-012 | |

*Appendix 1 Continued*

| Reagent type (species) or resource | Designation | Source or reference | Identifiers | Additional information |
|---|---|---|---|---|
| Commercial assay or kit | Pierce BCA Protein Assay Kit | Thermo Fisher Scientific | PI23225 | |
| Commercial assay or kit | EZQ Protein Quantitation Kit | Thermo Fisher Scientific | R33200 | |
| Commercial assay or kit | MidiMACS Starting Kit (LS) | Miltenyi Biotec | 130-042-301 | |
| Commercial assay or kit | Senescence Detection Kit | Abcam | Ab65351 | |
| Commercial assay or kit | Mouse XL Cytokine Array Kit | R&D Systems | ARY028 | |
| Commercial assay or kit | ADP/ATP Ratio Kit | Abcam | Ab65313 | |
| Commercial assay or kit | Novex Colloidal Blue Staining kit | Invitrogen | LC6025 | |
| Strain, strain background (*Mus musculus*) | C57BL/6J (2 months) male | Jackson Laboratory | IMSR_JAX:000664 | |
| Strain, strain background (*Mus musculus*) | C57BL/6 J (18 months) male | National Institute of Aging | N/A | |
| Strain, strain background (*Mus musculus*) | C57BL/6J (Nox4 knockout) male | Dr. Karl-Heinz Krause | University of Geneva | |
| Other | Mouse genotyping service | Transnetyx, Inc, Cordova, TN | | |
| Chemical compound, drug | Isoflurane | VetOne | NDC 13985-528-60 | Anesthesia agent |
| Other | Inverted microscope | Keyence | SCR_017202 | |
| Other | Inverted microscope | Carl Zeiss | | |
| Other | Nikon A1R confocal microscope | Nikon | | |
| Other | Microplate reader | BioTek | SynergyMx | |
| Other | Centrifuge | Eppendorf | 5702R | |
| Other | Flow Cell Sorter | BD Biosciences | | |
| Other | BD FACS Aria | BD Biosciences | SCR_018091 | |
| Other | Flow Cell Analysis | BD Biosciences | BD LSR II GUAVA EasyCyte | |
| Other | Extracellular Flux Analyzer (Seahorse XFe96) | Agilent | SCR_019545 | |
| Other | LS columns (magnetic separation) | Miltenyi Biotec | 130-042-401 | AEC2 purification |
| Other | Extracellular Flux Analyzer | Agilent | Seahorse XFe96 | |
| Software, algorithm | ImageJ | Wayne Rasband (NIH) | https://github.com/imagej/imagej1 | *Johannes, 2021* |
| Software, algorithm | Image Quant | Cytiva | Version TL8.1 | |

*Appendix 1 Continued on next page*

*Appendix 1 Continued*

| Reagent type (species) or resource | Designation | Source or reference | Identifiers | Additional information |
|---|---|---|---|---|
| Software, algorithm | FlowJo | Tree Star Inc | | |
| Software, algorithm | Nis Elements | Nikon | Version 5.0 | |
| Sequence-based reagent | Mouse Nox4 forward | IDT | ACT TTT CAT TGG GCG TCC TC | RT-PCR |
| Sequence-based reagent | Mouse Nox4 reverse | IDT | AGA ACT GGG TCC ACA GCA GA | RT-PCR |
| Sequence-based reagent | Mouse β-actin forward | IDT | AGT GTG ACG TTG ACA TCC GT | RT-PCR |
| Sequence-based reagent | Mouse β-actin reverse | IDT | TGC TAG GAG CCA GAG CAG TA | RT-PCR |

