## [Decision Letter]

**Acceptance summary:**

This paper is of interest to a broad audience of lung biology, aging biology, fibrosis and pulmonary medicine. It identified a critical mechanism under which aging of mesenchymal stromal cells deleteriously affect lung alveolar progenitor cell behavior. It employed 3-demensional organoid culture, proteomics, metabolic manipulation, as well as genetically modified mice to support their claims. They build on their previous discovery that oxidative stress is a key regulator of pulmonary fibrosis and that Nox4 is a critical regulator. These studies further support the concept of targeting oxidative stress through Nox4 as a viable therapeutic target in organ fibrosis.

**Decision letter after peer review:**

Thank you for submitting your article "Mesenchymal Stromal Cell Aging Impairs the Self-Organizing Capacity of Lung Alveolar Epithelial Stem Cells" for consideration by *eLife*. Your article has been reviewed by 2 peer reviewers, and the evaluation has been overseen by a Reviewing Editor and Paul Noble as the Senior Editor. The following individual involved in review of your submission has agreed to reveal their identity: Daniel Kass (Reviewer #2).

Essential revisions:

1) Please respond to the 2 reviewers suggestions to improved the manuscript. Some new data would be helpful where suggested.

*Reviewer #1:*

In this manuscript, Dr. Chanda, Dr. Thannickal, and colleagues investigated the role of lung mesenchymal stromal cell (L-MSC) aging in supporting type 2 alveolar epithelia cells (AEC2s) regeneration using 3D organoid culture. This study addresses very important aspects in aging related lung biology study. It is largely unknown the mechanistic links between niche aging and the stem cell exhaustion in the lung. The authors demonstrated that young L-MSCs supported the efficient alveoloshpere formation of both young and old AEC2s, while the aged L-MSC did not, even with young AEC2s. This study employed 3-dimensional organoid culture, proteomics, metabolic manipulation, as well as genetically modified mice to support their claim that aging of lung mesenchymal stromal cells leads to altered bioenergetics and oxidative stress that impair regenerative capacity of the epithelial stem cells. Some additional studies are suggested below to better strengthen the findings and provide more insight of cell-cell interaction between lung mesenchymal stromal cells type 2 alveolar epithelia cells.

1. A comparison of cytokines/growth factors or secreted proteins between aged WT L-MSCs and aged Nox4+/- L-MSCs would provide better clue to identify the secreted factors which is crucial for L-MSCs to support AEC2 organoid formation.

2. Adding H2O2 to the organoid culture of young WT L-MSCs/young AEC2s or aged Nox4+/- L-MSCs/young AEC2s would help to directly demonstrate whether Nox-4- dependent oxidative stress of L-MSCs is associated with impaired AEC2 organoid formation.

3. An experiment to test the effect of Nox-4 inhibitor on AEC2 organoid formation with Old WT L-MSCs would be informative.

4. In Figure 4G, it would be great is the authors include the culture of Nox-4- L-MSCs with old AEC2s to test if Nox-4- L-MSCs are also better in supporting old AEC2s renewal.

5. Some discussion should be added to explain how the metabolic dysregulation of aged L-MSCs links to the supportive function of the cells to AEC2 organoid formation.

*Reviewer #2:*

I have been asked to review by Drs Chanda and colleagues which reports a very interesting phenotype-the failure of aged lung mesenchymal stem cells to support alveolospheres. The team, which is well-suited to an analysis of senescence, found that aged lung MSCs exhibit altered bioenergetics compared to young L-MSCs. Altered energetics were associated with changes in the secretome of these cells reflecting many of the classical SASP. The aged MSCs express H2O2 which is associated with activation of Nox4. The team found that the phenotype on alveolospheres could be rescued with Nox4+/- aged MSCs. In general the manuscript is well-written and shares a very important message about the role of the mesenchymal compartment in regulating lung epithelial cell growth and differentiation.

The implications here are that an aging mesenchymal compartment in the lungs of older patients are unable to support epithelial cell repair. In the context of pulmonary fibrosis, these data also suggest that fibroblasts are not simply "matrix factories." They are critical to the maintenance of the gas exchange surface. The authors do not overstate their conclusions, which is scientifically appropriate but these data here do lead to some very exciting speculation about lung injury and repair.

The authors do show several bioenergetic and secretome changes associated with aging MSCs. These findings do not clearly establish what the critical factors are that support the formation of alveolospheres. This is a limitation of the manuscript. finding that factor or factors may represent finding a needle in a haystack. But the association of Nox4 haploinsufficiency with rescue of the phenotype does point the team in the right direction.

1. I wonder about some of the critical mesenchyme-derived factors that support lung development like FGF7/10 or Wnts, for example? Does the team have data on these factors? This manuscript suggests that fibroblast-free organoids can form in the presence of these factors: https://www.cell.com/iscience/pdf/S2589-0042(18)30252-9.pdf

2. Is there any epithelial cell cytotoxicity from the H2O2 generated by the aged stem cells? This may be difficult to tease out from general failure of epithelial cells to form organoids.

3. In Figure 1I, for example, the analysis should be two-way ANOVA to test a 2x2 experimental design where young/old MSCs and young/old epithelial cells are tested. The two way ANOVA calculation will provide an interaction term which may also be informative. This should apply to all 2x2 designs.

4. In the statistical analysis section, please identify the method for adjusting for false discovery rate.

---

## [Author Response]

Reviewer #1:In this manuscript, Dr. Chanda, Dr. Thannickal, and colleagues investigated the role of lung mesenchymal stromal cell (L-MSC) aging in supporting type 2 alveolar epithelia cells (AEC2s) regeneration using 3D organoid culture. This study addresses very important aspects in aging related lung biology study. It is largely unknown the mechanistic links between niche aging and the stem cell exhaustion in the lung. The authors demonstrated that young L-MSCs supported the efficient alveoloshpere formation of both young and old AEC2s, while the aged L-MSC did not, even with young AEC2s. This study employed 3-dimensional organoid culture, proteomics, metabolic manipulation, as well as genetically modified mice to support their claim that aging of lung mesenchymal stromal cells leads to altered bioenergetics and oxidative stress that impair regenerative capacity of the epithelial stem cells. Some additional studies are suggested below to better strengthen the findings and provide more insight of cell-cell interaction between lung mesenchymal stromal cells type 2 alveolar epithelia cells.

Thank you for the positive comments and recognition of the novelty of our studies. We have conducted additional studies as suggested and carefully revised the manuscript with changes highlighted in red, and a point-by-point response follows. We believe these additional studies further strengthen the findings and provide significant new insights into cell-cell interactions in the alveolar stem cell niche.

1. A comparison of cytokines/growth factors or secreted proteins between aged WT L-MSCs and aged Nox4+/- L-MSCs would provide better clue to identify the secreted factors which is crucial for L-MSCs to support AEC2 organoid formation.

Thank you for this suggestion. In new studies, we compared SASP-associated cytokines produced by aged wild-type and aged Nox4^-/+^ L-MSCs. Our results indicate that a number of secreted SASP cytokines are modulated by Nox4, and these data are now included in Figure 4—figure supplement 4I. We were unable to identify a group of cytokines (except for Coagulation Factor III) that were specifically identified in the dataset comparing young vs. aged L-MSCs based on the antibody-based cytokine array method. Further studies with an unbiased approach such as that employed by mass spectrometric analyses (Figure 2I) will be required for more in-depth analyses of Nox4 regulated SASP cytokines/growth factors. Additionally, we investigated the effect of Nox4 haploinsufficiency on cellular bioenergetics and senescence of L-MSCs. We observed significant reduction in real-time OCR, including basal, ATP-linked, proton leak, and maximal in aged Nox4-/+ L-MSCs as compared to aged L-MSCs (Figure 4G, H). β-gal activity, a marker of senescence, was also reduced in Nox4 deficient L-MSCs (Figure 4—figure supplement 4H).

2. Adding H2O2 to the organoid culture of young WT L-MSCs/young AEC2s or aged Nox4+/- L-MSCs/young AEC2s would help to directly demonstrate whether Nox-4- dependent oxidative stress of L-MSCs is associated with impaired AEC2 organoid formation.

Thank you. Although it is difficult to mimic the levels of continuous H_2_O_2_ production by aged L-MSCs in the alveolosphere model, we evaluated the effects of adding a single dose of H2O2 at a concentration of 1 μM to the organoid culture of young WT L-MSCs/young AEC2s. This resulted in a reduction in alveolosphere size, without affecting alveolar numbers (Figure 4—figure supplement 4, A-C).

3. An experiment to test the effect of Nox-4 inhibitor on AEC2 organoid formation with Old WT L-MSCs would be informative.

As suggested, we have performed new experiments testing the effect of the Nox4 inhibitor, GKT137831 on AEC2 organoid formation with WT aged L-MSCs. Interestingly, we were unable to detect significant differences in both alveolosphere size and number with GKT137831 treatment (Figure 4—figure supplement 4, K-M). It is important to recognize that we have not confirmed that this intervention effectively inhibited Nox4 activity (as this is more difficult to assess in the 3D organoid model). We believe the genetic approach we employed (Figure 4) provides more rigorous evidence that Nox4 effects are critical in niche support of AEC2 function.

4. In Figure 4G, it would be great is the authors include the culture of Nox-4- L-MSCs with old AEC2s to test if Nox-4- L-MSCs are also better in supporting old AEC2s renewal.

Thank you for this insightful suggestion. We did perform the suggested studies and found that aged L-MSCs Nox4^-/+^ were incapable of completely rescuing impaired alveolosphere formation when combined with aged AEC2s (Figure 4—figure supplement 4, N-P). The implications of this new data are discussed in the Results and Discussion section. While the phenotype of impaired alveolosphere formation in young AECs/aged MSCs was reversed by haploinsufficiency of Nox4 in the mesenchyme (Figure 4, I-K), this was not sufficient to reverse the phenotype in aged AECs. This finding, along with the observation that WT young L-MSCs are sufficient to reverse the aged AEC phenotype (Figure 1H) suggests that, in addition to reduced expression of Nox4 in aged L-MSCs, the young mesenchyme may support niche function and recovery of alveolosphere formation in aged AEC2s by a Nox4-independent mechanism. Thus, an aging alveolar stem cell niche may involve both a gain of “pro-aging factors” (Nox4) and loss of “rejuvenation factors” by the mesenchyme; the identification of the latter will require further study.

5. Some discussion should be added to explain how the metabolic dysregulation of aged L-MSCs links to the supportive function of the cells to AEC2 organoid formation.

Thank you. We have now added to the Results and Discussion (including the data above) with link to metabolic dysregulation (Figure 4G, H; Figure 4—figure supplement 4H).

Reviewer #2:1. I wonder about some of the critical mesenchyme-derived factors that support lung development like FGF7/10 or Wnts, for example? Does the team have data on these factors? This manuscript suggests that fibroblast-free organoids can form in the presence of these factors: https://www.cell.com/iscience/pdf/S2589-0042(18)30252-9.pdf

This is an interesting concept that highlights the complexity of stem cell niche maintenance. Our studies indicate that niche aging may involve both a GAIN of pro-senescent/aging factors and a LOSS of pro-regenerative factors. The relative contributions to these are unknown; however, we do show that suppressing pro-senescent factors (Nox4) was sufficient to recover alveolosphere formation in aged L-MSCs/young AEC2s (but not in aged L-MSCs/aged AEC2s). This may be related to the additional requirement for pro-regenerative (“rejuvenation”) factors in aged AEC2s. Based on the suggestion by the reviewer, we did perform studies with FGF-10 to determine whether alveolosphere formation can be rescued in aged L-MSCs/aged AEC2s. Although some experiments showed significant improvement in organoid formation, we did not get consistent data to be certain that replacing this factor (FGF-10) is sufficient. As discussed above, it is possible that modulation of both LOSS and GAIN components are required for optimal alveolosphere formation.

**Author response image 1. sa2fig1:** (**A**) Alveolosphere assay. Aged L-MSCs were co-cultured with aged AEC2s in the presence of FGF-10 (100 ng/ml). The alveolospheres were imaged by brightfield microscopy after 12 days of co-culture, and comparative outcomes are shown here (scale bar = 500 μm) (**B**) Alveolospheres in each well were counted. Alveolosphere size (volumes) were determined for each of the 2 co-culture groups. Nested scatterplot showing Mean ± SEM of all the alveolospheres counted in each well for each group (n = 2 mice; **p < 0.01; unpaired T-test). Data were natural log transformed before T-test.

2. Is there any epithelial cell cytotoxicity from the H2O2 generated by the aged stem cells? This may be difficult to tease out from general failure of epithelial cells to form organoids.

We agree. H_2_O_2_ may mediate more direct effects by inducing cytotoxicity or induce senescent changes such as activation of a SASP program. As noted in the response to Reviewer 1, addition of a low dose of H_2_O_2_ (1 μM) to the organoid culture of young WT L-MSCs/young AEC2s reduced alveolosphere size without a significant change in the number of alveolospheres formed (Figure 4—figure supplement 4, A-C).

3. In Figure 1I, for example, the analysis should be two-way ANOVA to test a 2x2 experimental design where young/old MSCs and young/old epithelial cells are tested. The two way ANOVA calculation will provide an interaction term which may also be informative. This should apply to all 2x2 designs.

Thank you, and we agree. The analysis results presented in Figure 1I and 4J were obtained by performing two-factor ANOVA testing groups including young/aged L-MSCs and young/aged epithelial cells (AEC2s). Statistical analysis section in the manuscript has been revised to clarify these changes.

4. In the statistical analysis section, please identify the method for adjusting for false discovery rate.

The statistical analysis section of the manuscript has been revised to include a description of the method, Tukey’s correction, used for adjusting Type I error for pairwise comparisons.